# The Role of the Microbiome in Pancreatic Cancer

**DOI:** 10.3390/cancers14184479

**Published:** 2022-09-15

**Authors:** Koji Miyabayashi, Hideaki Ijichi, Mitsuhiro Fujishiro

**Affiliations:** Department of Gastroenterology, Graduate School of Medicine, The University of Tokyo, Tokyo 113-8655, Japan

**Keywords:** pancreatic ductal adenocarcinoma (PDAC), microbiome, tumor microenvironment, immune checkpoint blockade (ICB), intraductal papillary mucinous neoplasm (IPMN), chronic pancreatitis (CP)

## Abstract

**Simple Summary:**

Pancreatic cancer is deadly cancer characterized by dense stroma creating an immunosuppressive tumor microenvironment. Accumulating evidences indicate that the microbiome plays an important role in pancreatic cancer development and progression via the local and systemic inflammation and immune responses. The alteration of the microbiome modulates the tumor microenvironment and immune system in pancreatic cancer, which affects the efficacy of chemotherapies including immune-targeted therapies. Understanding the role of microbiome and underlying mechanisms may lead to novel biomarkers and therapeutic strategies for pancreatic cancer. This review summarizes the current evidence on the role of the microbiome in pancreatic cancer.

**Abstract:**

Pancreatic ductal adenocarcinoma (PDAC) is one of the most lethal malignancies, with little improvement in outcomes in recent decades, although the molecular and phenotypic characterization of PDAC has contributed to advances in tailored therapies. PDAC is characterized by dense stroma surrounding tumor cells, which limits the efficacy of treatment due to the creation of a physical barrier and immunosuppressive environment. Emerging evidence regarding the microbiome in PDAC implies its potential role in the initiation and progression of PDAC. However, the underlying mechanisms of how the microbiome affects the local tumor microenvironment (TME) as well as the systemic immune system have not been elucidated in PDAC. In addition, therapeutic strategies based on the microbiome have not been established. In this review, we summarize the current evidence regarding the role of the microbiome in the development of PDAC and discuss a possible role for the microbiome in the early detection of PDAC in relation to premalignant pancreatic diseases, such as chronic pancreatitis and intraductal papillary mucinous neoplasm (IPMN). In addition, we discuss the potential role of the microbiome in the treatment of PDAC, especially in immunotherapy, although the biomarkers used to predict the efficacy of immunotherapy in PDAC are still unknown. A comprehensive understanding of tumor-associated immune responses, including those involving the microbiome, holds promise for new treatments in PDAC.

## 1. Introduction

Pancreatic ductal adenocarcinoma (PDAC) is a deadly cancer worldwide, and it has a five-year survival rate of less than 9% for all the stages combined [1]. More than half of PDAC patients are diagnosed as inoperable with metastatic diseases or advanced diseases. PDAC frequently recurs even after resection, and chemotherapies are frequently ineffective. Early diagnosis methods and new therapeutic strategies are needed.

According to the development of the genetic and molecular characterization of PDAC, tailored therapies have emerged. Recent studies have suggested that up to 25% of PDACs have actionable genetic mutations [2] and three subgroups of PDAC patients are considered to be possibly targeted by tailored therapies. Patients with gene alterations of homologous recombination deficiency (HRD), such as *BRCA1* and *BRCA2* mutations, benefit from platinum-based therapy and poly (ADP-ribose) polymerase (PARP) inhibitors [3,4,5,6,7,8]. Patients with mismatch repair deficiency (MMR-D), including high microsatellite instability (MSI-H) and high tumor gene mutation burden (TMB-H), can be targeted by immune checkpoint blockade (ICB) therapies [2,9]. Patients with wild-type *KRAS* (*KRASWT*) often have alternative oncogenic mutations, such as *BRAF* [3,5], and may be candidates for small-molecule therapies.

Although ICB is a promising therapy in many types of cancers as well as PDAC, biomarkers for the efficacy of treatment are still unknown in PDAC. Clinical trials showed that MMR-D patients in PDAC are resistant to ICB therapies compared to other types of cancer [2,9]. PDAC is characterized by a dense stromal component that interacts with cancer cells and serves as a tumor-supportive environment [10]. Tumor-infiltrating T cells play an important role in eliminating tumor cells, and these components are regulated by other types of cells in tumor microenvironment (TME) such as fibroblasts, macrophages, and dendritic cells [11]. Cancer cells orchestrate stromal cells to create an immunosuppressive environment that is favorable to cancer cells. Dense stroma creating an immunosuppressive TME is one of the reasons for the complexity of chemoresistance in PDAC. Intrinsic factors in PDAC cells as well as extrinsic factors in non-cancer cells are associated with the formation of immunosuppressive TME. A comprehensive analysis of PDAC using multi-omics studies, including the microbiome, would contribute to the understanding of complexed TME and the establishment of new therapeutic strategies.

The microbiome is now known to be associated with cancer development and progression in many types of cancer [12]. Several studies have revealed the association between PDAC progression and the oral, gut, and intratumor microbiomes, although the identified bacteria differ between reports [13]. These reports have commonly reported that high microbial diversity is associated with favorable outcomes. Bacteria are thought to migrate from the gut to the pancreas, and a recent report has suggested that the gut microbiota modifies the overall microbiome of tumors [14,15]. Regarding the early detection of PDAC, evidence is accumulating to suggest that the microbiome is associated with premalignant diseases of the pancreas, such as chronic pancreatitis (CP) and intraductal papillary mucinous neoplasia (IPMN) [16,17]. Although further studies are needed to understand whether the differences in the microbiome in pancreatic precursors of PDAC are a cause or a consequence, the microbiome has a potential role in the early detection of PDAC. Regarding the treatment of PDAC, intratumor CD8 + T cell infiltration may play an important role in microbiome-associated immune modification [15]. The microbiome can be a biomarker of the efficacy of ICB therapies. Furthermore, antibiotic treatment may provide new options to modify the efficacy of chemotherapies as well as ICB therapies.

In this review, we summarize the role of the microbiome in the development and progression of PDAC and discuss the potential role of the microbiome in the design of new therapeutic strategies, especially in immunotherapy. A comprehensive understanding of the microbiome-associated immune response in tumors holds promise for new treatments in PDAC.

## 2. Association of Oral, Gut, and Intratumor Microbiomes with PDAC

For many types of cancer, the microbiome is now recognized to be involved in the development and progression of cancer [12], including *Helicobacter pylori* in gastric cancer [18] and *Enterotoxigenic Bacteroides* fragilis [19] and *Fusobacterium nucleatum* [20,21] in colon cancer. Even in cancers other than those of the digestive tract, such as breast cancer, gut dysbiosis can alter the host’s inflammatory response, promote systemic fibrosis and collagen deposition, and contribute to the metastasis of cancer cells [22].

### 2.1. Association of Oral and Gut Microbiomes in PDAC

In recent years, many papers have reported the association of microbiome and PDAC (Figure 1). The microbiome of the pancreas is reflected by a retrograde bacterial migration from the duodenum.

Many similar observational results have been reported comparing the microbiome between PDAC patients and healthy controls using oral and fecal samples. Farrell et al. [23] analyzed oral microbiota and reported that the levels of *Neisseria elongata* and *Streptococcus mitis* were significantly reduced in patients with PDAC relative to healthy controls. Fan et al. [24] reported that *Porphyromonas gingivalis* and *Aggregatibacter actinomycetemcomitans* were associated with a higher risk of PDAC. Michaud et al. (2013) reported that high levels of antibodies against the oral bacteria *Porphyromonas gingivalis* were also associated with a twofold increased risk of PDAC [25]. In addition, the authors commented that the antibody of *Porphyromonas gingivalis* in plasma was measured up to 10 years prior to PDAC diagnosis, which suggested a potential role for the measurement of plasma antibody levels of *Porphyromonas gingivalis* in the early detection of PDAC. Fecal metagenomic classifiers [26] were reported, and the authors showed that non-invasive, robust, and specific fecal microbiota-based screening is effective for the early detection of PDAC. Half et al. [27] analyzed the fecal microbiota of patients with PDAC as well as pre-cancerous lesions and healthy controls and observed an increase in *Bacteroidetes* and a decrease in *Firmicutes* in PDAC patients. At taxonomic levels, *Anaerostipes* and genera belonging to *Erysipelotrichaeceae* and *Clostridiaceae* decreased in PDAC, while genera belonging to *Veillonellaceae* increased in PDAC. Pushalkar et al. [15] analyzed gut microbiota and reported that *Proteobacteria* constituted more than half of the gut microbiome in PDAC patients, while it comprised only 8% of the bacteria in normal pancreata. The authors reported that *Synergistetes* and *Euyarchaeota* were similarly enriched in the gut microbiome in patients with PDAC. In addition, smoking is a well-described risk factor for PDAC, and smoking cessation results in an increase in *Firmicutes* and *Actinobacteria* and a decrease in *Proteobacteria* and *Bacteroidetes* species within the intestine [28]. A recent report from Japan showed significant enrichments of *Streptococcus* and *Veillonella* spp and a depletion of *Faecalibacterium prausnitzii* in oral and fecal samples in PDAC patients, which are commonly observed in German and Spanish cohorts [29].

### 2.2. Association of Intratumor Microbiomes in PDAC

Geller et al. [30] analyzed bacterial DNA in PDAC tissues compared to normal tissues and an increased presence of bacterial DNA was observed, with *Gammaproteobacteria* being the most abundant in PDAC.

Nejman et al. [31] analyzed the intratumor microbiome across seven cancer types compared to adjacent normal tissues and reported that *Proteobacteria* dominated the microbiome of PDAC, similarly to the normal duodenal microbiome. *Fusobacterium* was more abundant in gastrointestinal (GI) cancers, such as colorectal cancers, stomach cancers, cholangiocarcinoma and PDAC, compared to non-GI cancers [31,32]. *Fusobacterium nucleatum* was enriched in PDAC as well as breast cancer. Interestingly, a distinct microbiome was observed according to subtypes of breast cancer.

Riquelme et al. [14] showed that a diverse intratumor microbiome signature enriched with *Pseudoxanthomonas*, *Streptomyces*, *Sacchropolyspora*, and *Bacillus clausii* was associated with favorable survival in multiple patient cohorts, while no predominant microbiome was identified in patients with short survival [14].

Consequently, dysbiosis of the oral, gut, and intratumor microbiomes is associated with the development of PDAC, and microbe-induced inflammation modifies a systemic immune response, which supports the progression of PDAC [14,15]. The oral, gut, and intratumor microbiomes of human PDAC patients also harbors specific bacteria compared to those with normal pancreata, suggesting the role of the microbiota in regulating PDAC progression. Together, these results suggest that the microbiota is not only directly involved in oncogenesis locally within the TME but also triggers a systemic immune response that contributes to the development of PDAC.

## 3. Association of Microbiomes with Pancreatic Diseases at High Risk for PDAC

Early detection of PDAC is an important strategy to improve outcomes because PDAC is highly resistant to conventional chemotherapies and molecularly targeted reagents. Chronic pancreatitis and pancreatic cysts, especially IPMN, are recognized as risk factors for PDAC as well as smoking, diabetes, obesity, and a family history of PDAC. Efforts have been made to find biomarkers that distinguish between precursor diseases and PDACs, but efficient biomarkers are still unknown. Accumulating evidence indicates the association between microbiota and PDAC as well as the potential role of microbiota as a biomarker for early detection of PDAC.

### 3.1. Association of Microbiomes with Chronic Pancreatitis

Chronic pancreatitis (CP) is a disease of the pancreas in which ongoing inflammation leads to atrophy and fibrosis of the pancreas and loss of endocrine and exocrine secretion. Possible causes include toxic factors such as alcohol, smoking, diabetes mellitus, idiopathic, heredity, autoimmune reactions, and obstructive mechanisms. CP can cause recurrent abdominal pain, diabetes mellitus (endocrine dysfunction), and indigestion (exocrine dysfunction). Fibrosis and calcification of the pancreatic parenchyma as well as dilatation of the pancreatic ducts were observed at the late stage, and CP is known to be a risk for PDAC. It is important to understand the biology of CP and mechanisms of carcinogenesis from CP patients as well as to discover biomarkers for the early detection of PDAC in CP patients.

Recent analyses compared the gut microbiome in patients with CP and healthy controls (HC). Ciocan et al. [16] analyzed intestinal microbiota profiles in severe alcoholic hepatitis (sAH) or alcoholic chronic pancreatitis (ACP) compared to alcoholic healthy controls (A-HC). The authors reported that patients with ACP have lower bacterial diversity compared to that of A-HC. A more active intestinal microbiome was observed in patients with ACP (e.g., *Klebsiella*, *Enterococcus,* and *Sphingomonas*). A lower abundance *of Faecalibacterium* was observed in ACP compared to both A-HC and sAH patients. A decrease in the abundance of *Faecalibacterium prausnitzii* was also observed in patients with CP [33]. Jandhyala et al. [33] evaluated the intestinal microbiota in 30 patients with CP and 10 HC. The authors reported a reduction in the abundance of *Faecalibacterium prausnitzii* and *Ruminococcus bromii* from controls compared to those with CP. *Faecalibacterium prausnitzii* is the most abundant commensal in the human intestine [34], which is reported to be associated with epithelial barrier function due to stimulation of mucin production and tight-junction molecules [35,36,37]. *Ruminococcus bromii* is associated with starch degradation harvesting butyrate in the human colon [38]. Therefore, a decrease in *Faecalibacterium prausnitzii* and *Ruminococcus bromii* may contribute to the disruption of the intestinal mucosal barrier. Zhou et al. [39] reported that patients with CP showed dysbiosis of the gut microbiota with decreased diversity. A low abundance of Firmicutes and Actinobacteria and high abundance of *Proteobacteria* were observed in CP patients compared to the HC group. A high abundance of *Escherichia*/*Shigella* and a low abundance of *Faecalibacterium* were observed in CP patients. In these studies, the microbial diversity was decreased in patients with CP compared to HC [16,33,39]. Specific organisms, such as Bacteroidetes and *Faecalibacterium,* were decreased in CP compared to HC [16,33], and *Proteobacteria* was increased in CP compared to HC [16,39]. In addition, the endotoxin level was elevated in patients with CP compared to HC [33,39].

These observations suggest the association of the microbiome with CP. However, it remains unclear whether dysbiosis in CP is a cause or consequence of pancreatitis.

Pancreatic exocrine insufficiency and diabetes mellitus have been associated with the incidence of CP. The abundance of *Faecalibacterium prausnitzii* was negatively correlated with glycemic control [33,39]. Bifidobacterium was negatively correlated with pancreatic exocrine insufficiency [33,39]. These results suggest that changes in the microbiome are associated with the severity of CP.

Regarding autoimmune pancreatitis (AIP), which is another type of CP, Hamada et al. [40] reported that the proportions of Bacteroides, Streptococcus, and Clostridium species were higher in patients with CP compared to AIP. Nishiyama et al. [41] reported that an increased abundance of *Akkermansia muciniphila* I and *Lactobacillus reuteri*, which are beneficial bacteria, was observed after the administration of pancreatic digestive enzymes in mice [41], suggesting that the nutritional status affects the gut microbiota. These results suggest that dysbiosis of the gut microbiome due to pancreatic exocrine insufficiency and malabsorption commonly exists in both AIP and CP.

As mentioned previously, CP is a risk of PDAC, and the discovery of new biomarkers for early detection of PDAC is needed. Farrell et al. [23] reported that the levels of *Neisseria elongata* and *Streptococcus mitis* were significantly decreased in patients with PDAC compared to HC. Additionally, the levels of one increased species (i.e., *Granulicatella adiacens*) and one decreased species (*Streptococcus mitis*) were significantly different between PDAC and CP. As such, the microbiome provides the potential to detect early cancer in CP patients (Table 1).

### 3.2. Association of Microbiomes with Precursor Diseases of PDAC

Intraductal papillary mucinous neoplasms (IPMNs) of the pancreas have been recognized as precursor lesions to PDAC. IPMNs have been categorized into main-duct and branch-duct types according to the location of the lesions, such as cystic dilatation of the ducts. As main-duct-type IPMNs have high malignant potential, surgical resection is recommended at the time of diagnosis. In patients with branch-type IPMNs, the size of the cystic lesions and the diameter of the main pancreatic duct were associated with the incidence of IPMN-derived carcinoma, whereas they are not associated with the concomitant occurrence of PDAC [42]. Similarly, in this precursor, there are still no clinically efficient biomarkers for the early detection of PDAC. The correlation of IPMN and the microbiome has been reported [17,43], and the microbiome has the potential to be a marker for the early detection of PDAC (Table 1).

Previously, Li et al. [43] analyzed the microbiomes of 33 patients with pancreatic cystic fluid, including IPMN, mucinous cystic neoplasm (MCN), pseudocysts, and serous cystic neoplasm (SCN). The authors reported the presence of *Bacteroides*, *Escherichia*/*Shigella*, and *Acidaminococcus* as predominant genera and did not find significant differences in the diversity of bacterial microbiota between the cyst types. The authors indicated the unique bacterial ecosystem in pancreatic cyst fluid and identified 17 potentially pro-cancerous bacteria including Bifidobacterium, *Faecalibacterium*, *Escherichia/Shigella*, and *Bacteroides*. Olson et al. [44] found no differences in the alpha diversity of the oral microbiota between patients with PDAC and HC, or between patients with PDAC and those with IPMNs. The authors observed that the PDACs had higher levels of *Firmicutes* and HC had higher levels of *Proteobacteria*. The differences between patients with PDACs and IPMNs were similar with those between PDACs and controls. This may be due to the small difference between PDACs and IPMNs and the smaller sample size of IPMNs than controls.

Recently, Gaiser et al. [17] analyzed the microbiota in 105 patients with pancreatic cyst fluid including IPMN, IPMN with cancer, SCN, and MCN. The authors reported that significantly higher bacterial DNA copies were found in the cyst fluid of IPMN and cancer compared with non-IPMN (SCN and MCN). Microbiome analysis of cyst fluid samples from IPMN with low-grade dysplasia (IPMN LGD) or IPMN with high-grade dysplasia (IPMN HGD) and cancer revealed that the three groups showed no significant differences for diversity at the operational taxonomic unit (OTU) level. At the phylum level, IPMN LGD was found to be dominated by *Proteobacteria*, while IPMN HGD and cancer were generally diverse and dominated by *Firmicutes* or *Proteobacteria*. Despite the large individual variation, the authors found that IPMN HGD was enriched in *Fusobacteria*, *Granulicatella*, and *Serratia*, suggesting the potential role of these bacteria in the progression of pancreatic precursors to PDAC. Kohi et al. [45] analyzed duodenal microbiota in patients with PDAC and pancreatic cysts. The authors reported that patients with PDAC had significantly decreased alpha diversity in duodenal microbiome compared to HC and pancreatic cysts. A high abundance of Bifidobacterium genera was observed in the duodenal fluid of PDAC patients compared to HC and pancreatic cysts. *Fusobacteria* and *Rothia bacteria* in duodenal fluid were enriched in PDAC patients with short survival. However, the authors reported that the microbiome profiles in duodenal fluid were not significantly different between HC and pancreatic cysts.

**Table 1 cancers-14-04479-t001:** The enriched microbiomes in oral, gut, and, intratumor in CP and IPMN.

Disease	Study Population	Specimen	Key Finding	Reference
PDAC and IPMN	Human	Pancreatic cystic fluid (Surgery)	Metabolites of potential bacterial origin (conjugated bile acids, free and carnitine-conjugated fatty acids, and TMAO) in cyst fluid were identified.	Morgel [46]
PDAC, pancreatic cysts, and normal	Human	Duodenal fluid	Duodenal fluid microbiome profiles were not significantly different between control subjects and patients with pancreatic cyst(s). Bifidobacterium genera was enriched in PDAC patients compared to control subjects and patients with pancreatic cyst(s).	Kohi [45]
IPMN and non-IPMN pancreatic cysts	Human	Pancreatic cystic fluid (Surgery)	Intracystic bacterial 16S DNA copy number and IL-1β protein quantity were significantly higher in IPMN with high-grade dysplasia and IPMN with cancer compared with non-IPMN PCNs. Fusobacterium nucleatum and Granulicatella adiacens in cyst fluid from IPMN with high-grade dysplasia	Gaiser [17]
IPMN, MCN, SCN, and normal	Human	Pancreatic cystic fluid (FNA)	Bacteroides spp., Escherichia/Shigella spp., and Acidaminococcus spp. which were predominant in PCF, while also a substantial Staphylococcus spp. and Fusobacterium spp. component was detected.	Li [43]
PDAC, IPMN, and normal	Human	Saliva	Firmicutes was relatively enriched and Proteobacteria is relatively decseaed in PADC compared to normal and IPMNs. No differences in diversity between patients with PDAC and healthy controls, or between patients with PDAC and those with IPMNs.	Olson [44]
CP and normal	Human	Fecal samples	Gut microbiota dysbiosis with decreased diversity was observed inpatients with CP. Firmicutes and Actinobacteria were decreased and Proteobacteria was enriched in CP group compared to HC group. Escherichia-Shigella was high and Faecalibacterium was low in CP group.	Zhou [39]
Alcoholic CP and alcoholic control	Human	Fecal samples	Bacterial diversity was lower in patients with ACP than that of AC. 17 genera differed betweem ACP and HC group. Klebsiella, Enterococcus and Sphingomonas were more frequent in patients with ACP.	Ciocan [16]
AIP and CP	Human	Fecal samples	Bacteroides, Streptococcus and Clostridium species were enriched in patients with CP.	Hamada [40]
CP with and without Diabetes	Human	Fecal samples	Plasma endotoxin concentrations was increased from controls to CP non-diabetics to CP diabetics. Alpha diversity between the groups were significantly different. Firmicutes:Bacteroidetes ratio was increased in CP patients without and with diabetes. Faecalibacterium prausnitzii and Ruminococcus bromii was decreased from controls to CP non-diabetics to CP diabetics.	Jandhyala [33]
PDAC, CP and normal	Human	Saliva	Neisseria elongata and Streptococcus mitis showed significant variation between patients with pancreatic cancer and controls. Granulicatella adiacens and S mitis showed significant variation between chronic pancreatitis samples and controls.	Farrell [23]

Morgell et al. [46] identified metabolites of potential bacterial origin including conjugated bile acids, free and carnitine-conjugated fatty acids, and trimethylamine N-oxide (TMAO) in pancreatic cyst fluid and showed that the levels of these metabolites correlated with the abundance of bacteria in the cyst using 16S gene analysis.

Although there are many reports suggesting a correlation between IPMN and the microbiome, further studies are needed to detect a relevant biomarker for the early detection of PDAC from pancreatic precursor diseases. Combining a multi-omics approach with microbiome analysis may lead to clinically accessible methods for the early detection of PDAC. Furthermore, the role of the microbiome in tumorigenesis is still unknown. As seen in the analysis of PDAC, microbiome-induced inflammation or TME modulation may contribute to the tumorigenesis and progression of pre-cancerous lesions to PDAC.

## 4. Mechanisms of Role of Microbiomes in PDAC

Regarding the underlying mechanisms, a major role of microbiota in PDAC development may be microbiome-induced inflammation and modification of the immune program. PDAC is characterized by an abundant stroma with desmoplasia creating an immunosuppressive environment. The desmoplasia, which creates a physical barrier around the tumor cells and prevents appropriate vascularization and delivery of chemotherapeutic agents, was previously thought to promote cancer [47], and many clinical trials were conducted to target it. However, most of those trials failed, and the stroma is now thought to be multifaceted [48,49,50]. Single-cell RNA sequencing also revealed subtypes of cancer-associated fibroblasts (CAFs) [51,52,53,54], which are associated with the subtypes of cancer cells. These subtypes of CAFs and cancer cells are associated with the infiltration of immune cells in tumors and the formation of an immunosuppressive tumor microenvironment.

Accumulating evidence suggests that the microbiome modulates innate and adaptive immune programs and contributes to the formation of an immunosuppressive tumor microenvironment. Understanding the underlying mechanisms of how microbiota affect the tumor microenvironment can lead to new therapeutic strategies.

### 4.1. Association of Microbiomes with Molecular Subtypes of Cancer Cell

In recent years, intense genomic analyses have been performed to reveal the mutational landscape of PDAC [55,56,57,58]. The frequently reported genetic mutations are concentrated in core signaling pathways including KRAS, WNT, NOTCH, DNA damage repair, RNA processing, cell-cycle regulation, transforming growth factor beta (TGF-β) signaling, switch or sucrose non-fermentable (SWI/SNF), chromatin regulation, and axonal guidance [55,56,57,58]. Recent comprehensive sequencing analysis elucidated transcriptional molecular subtypes of the cancer cells of PDAC including basal-like or squamous and classical or progenitor subtypes. Basal-like or squamous tumors are associated with poor outcomes and treatment resistance compared to classical or progenitor subtypes [58,59,60,61,62,63,64,65,66,67]. In addition to genome-based precision medicine, tailored therapies based on transcriptomic subtypes have emerged. Recent clinical trials have revealed that basal-like tumors are resistant to FOLFIRINOX-based therapies [67,68]. These results were supported by a study using patient-derived organoids (PDOs) by Tiriac et al. [69], who showed that chemotherapy signatures based on PDO could predict the treatment response in PDAC patients. Although the underlying mechanisms of this chemoresistance in basal-like tumors are still unknown, subgroups in basal-like subtypes characterized by the activation of KRAS, MYC, ∆N isoform of TP63 (∆Np63), and GLI2 [58,60,61,70,71,72,73,74,75] may be the key to solving the problem. As mentioned previously, these subtypes are associated with stromal subtypes including tumor-infiltrating lymphocytes (TILs) that are associated with the response to ICB therapies. Bailey et al. [58] identified an immunogenic subtype of cancer cells, which seems to be related to the immune-rich subtype of Maurer et al. [63]; these subtypes are associated with significant immune-cell infiltration and might hold promise as a biomarker for immunotherapy.

An association between the molecular subtypes and microbiome was recently reported, identifying a high abundance of *Acinetobacter*, *Pseudomonas,* and *Sphingopyxis* in basal-like human PDAC, and the analysis of microbial genes suggested the potential of the microbiome in inducing pathogen-related inflammation [76].

### 4.2. Role of Microbiomes in TME

In recent studies, single-cell RNA sequencing has been used to reveal the heterogeneity of stromal components [51,52,53,54]. CAFs play an important role in the regulation of the TME, and it has been reported that cancer-derived IL-1 or TGF-β can differentiate surrounding fibroblasts into inflammatory and myofibroblastic CAFs, respectively [52]. IL-6 secreted by inflammatory CAFs promote tumor growth, while myofibroblastic CAFs produce surrounding stroma. Since cancer cells create a microenvironment favorable to themselves, these stromal subtypes are related to the cancer-cell subtypes described above. Maurer et al. [63] reported CAF subtypes by RNA sequencing separately harvested PDAC epithelium and adjacent stroma using laser capture microdissection. The authors identified two subtypes that reflect ECM deposition and remodeling (ECM-rich) versus immune-related processes (immune-rich). ECM-rich stroma was strongly associated with basal-like tumors, while immune-rich stroma was more frequently associated with classical tumors [63,77]. Thus, the cancer cell subtypes and stromal subtypes were partially related, suggesting that they can be potential biomarkers for therapies targeting stroma in PDAC.

Dense stroma with desmoplastic reaction may act as a physical barrier and affect the infiltration of MDSCs and T cells in TME [78,79]. In addition, PDAC shows substantial immunological heterogeneity influencing T-cell infiltration [80,81,82,83,84,85,86], the level of T-cell infiltration is important in predicting the efficacy of ICB therapies, and patients with MSI-H tumors show abundant TILs and sensitivity to immune-targeted therapies [87,88,89,90,91,92,93]. Studies in mouse models have revealed potential targets, such as colony-stimulating factor 1 receptor, cytotoxic T-lymphocyte-associated protein 4 [94,95], and CXC chemokine receptor 2 [96,97] in combination with ICB, which have been tested in clinical trials. These results suggest that both the quality and quantity of CD8 + T cells in tumors are important in predicting the efficacy of immunotherapy, and that new biomarkers are needed to predict the status of infiltrating CD8 + T cells in tumors.

As an association between microbiome and CAF subtypes was not clear, the microbiome was associated with the inflammatory and immunosuppressive TME in mice.

The association of the mycobiome with the complement system has been reported [98]. The mycobiome promoted tumor growth due to mannose-binding lectin(MBL)–C3 cascade in a genetically engineered mouse model (GEMM) [98]. The complement system is an important component of the inflammatory response, which is involved in tumorigenesis and the adaptive immune response, which modulates T cell activation. The mycobiome has also been reported to enhance oncogenic KrasG12D-induced IL-33 secretion from PDAC and activates TH2 and ILC2 cells, which contribute to tumor progression using GEMM [99]. Anti-IL-33 or anti-fungal treatment decreases TH2 and ILC2 infiltration and increases the survival in GEMM.

Microbiota-induced activation of toll-like receptors (TLRs), especially TLR9, activate pancreatic cancer stellate cells and attract immunosuppressive T regulatory cells and MDSCs to the tumor environment, which contribute to the suppression of innate and adaptive immunity in PDAC progression in mice [100]. Lipopolysaccharide and TLR4 ligation induce a dendritic-cell-dependent immune response in the pancreas and increase pancreatic tumorigenesis, where Myd88 inhibition induced fibroinflammation via dendritic cells andTh2-derived CD4 T cells [101]. In addition, microbiota-mediated TLR2 and TLR5 ligation alters macrophages into an immunosuppressive phenotype and suppresses the T-cell-mediated antitumor immune response in mice [15]. Furthermore, microbial metabolism and metabolites can alter the TME. Obesity alters the gut microbiota and increases the level of the microbial metabolite deoxycholic acid (DCA), which induces DNA damage in obesity-associated hepatocellular carcinoma development in mice [102]. This metabolite may also be a risk factor for obesity-induced PDAC. Hezaveh et al. [103] showed that the aryl hydrocarbon receptor (AhR), which is a sensor of products of tryptophan metabolism, modulates immunity due to tumor-associated macrophage (TAM) function in murine PDAC. TAMs AhR activity was dependent on Lactobacillus metabolization of dietary tryptophan to indoles. Inhibition of AhR in myeloid cells reduced PDAC growth due to increased infiltration of IFNγ + CD8 + T cells in murine PDAC tumor [103]. Moreover, cytidine deaminase, an enzyme expressed by many bacteria, converts active gemcitabine into an inactive metabolite in colon cancer mouse models. *Gamma Proteobacteria* are reported to present in PDAC tumors and induce resistance to gemcitabine via cytidine deaminase [30]. Therefore, when antibiotics are used to reduce the bacteria, resistance to gemcitabine is eliminated. Thus, microbiome-based therapy may be useful not only for the suppression of carcinogenesis but also for preventing resistance to treatment.

Thus, the microbiome plays a pro-tumorigenic role via inflammation, immune response, and metabolic pathways (Figure 2). These results suggest the potential of the microbiome as a biomarker in immunotherapy and microbiome-targeted therapies.

## 5. Role of Microbiomes in PDAC Treatment

The curative therapy for localized PDAC is surgical resection with neoadjuvant and/or adjuvant therapies, but only approximately 20% of PDAC cases are resectable at diagnosis. For advanced PDAC, FOLFIRINOX and gemcitabine plus nab-paclitaxel [104,105] are the current standard of care. These two regimens have improved progression-free survival (PFS) and overall survival (OS) in metastatic PDAC, but most patients eventually progress and receive second-line chemotherapy. Recent improvements in genetic analysis technology have led to the realization of personalized medicine using genetic information. Currently, clinical trials are being conducted to tailor treatment to the underlying mutations [3,4,5]. In addition, a variety of immune-targeted approaches have emerged and showed limited success in disease control and survival [106]. The microbiome has the potential to be involved in the decisions of therapeutic strategies in PDAC because the microbiome is associated with chemoresistance and an immunosuppressive TME, which affect the efficacy of immune-related therapies.

### 5.1. Current Immunotherapy in PDAC

ICB therapies represent an effective treatment for patients with MMR-D/MSI-H regardless of the tumor type, but their activity depends on the tumor type. MMR-D is caused by the loss of function of MMR genes (*MLH1*, *PMS2*, *MSH2*, *MSH6*) due to hereditary germline mutations, known as Lynch syndrome, and biallelic somatic mutations of MMR genes. MMR-D and MSI-H are commonly associated with a high tumor gene mutation burden (TMB). High TMBs are thought to increase the number of neoantigens that are recognized by the host immune system, and activated tumor-infiltrating lymphocytes (TILs), especially CD8 + T cells, migrate into TME and play an important role in antitumor response [80,84].

The degree of T-cell infiltration in tumors is critical for predicting the efficacy of ICB therapy in other types of cancers [87,88,89,90,91,92], and a small subset of patients with MSI-H tumors exhibit T-cell infiltration and sensitivity to immunotherapy [93]. In PDAC, only 1% of patients have MMR-D or MSI-H [107,108]. In addition, ICB therapy is less effective in PDAC compared to other cancer types in the KEYNOTE 158 study [9] and NCI-MATCH study [2]. These results suggest that ICB responses depend on cancer-type-specific biological factors, even in patients with MMR-D. Furthermore, some MMR gene mutations may be passenger mutations, and responses to ICB therapy may be influenced by founder mutations that are important for the molecular behavior of cancer [109]. In case these markers are highly associated with MMR-D-driven tumorigenesis, MSI-H and high TMB may be a biomarker of the ICB response [110]. Further prospective studies are needed to discover a biomarker to predict intratumor T-cell infiltration and the ICB response in PDAC. In addition to ICB therapies, a variety of immune-targeted approaches have been tested in clinical trials with PDAC patients including tumor vaccines [111], such as PEGylated interleukin (IL)-10 [112] and GVAX, granulocyte-macrophage colony-stimulating factor (GM-CSF)-transfected tumor cells [113], and CAR(T) therapy, which showed limited success [106].

### 5.2. Role of Microbiomes as Biomarkers for Immunotherapy and Chemotherapy in PDAC

The association between the microbiome and the response of immunotherapy was reported in other types of cancer [114,115,116]. In these studies, the fecal microbiota of responders and non-responders to ICB therapy were compared. They reconstituted the germ-free (GF) mice with stool from the responders and specific candidate bacteria and succeeded in recreating the response to immunotherapy. Matson et al. [116] identified *Bifidobacteriaceae*, *Collinsella aerofaciens*, and *Enterococcus faecium*, and *Gopalakrishnan* et al. [115] identified *Fecalibacterium* spp. in melanoma patients. Routy et al. [114] identified Akkermansia muciniphila in lung cancer, renal cancer, and bladder cancer. In PDAC, Pushalkar et al. [15] recently reported that the depletion of the gut microbiome enhances the effect of ICB therapy. A recent study by Riquelme and colleagues [14] showed that the composition of the pancreatic tumor microbiome influences patient survival. In particular, a diverse intratumor microbiome signature enriched with *Pseudoxanthomonas*, Streptomyces, *Sacchropolyspora*, and *Bacillus clausii* predicted long-term survival in multiple patient cohorts [14]. It is very interesting to note that these studies identified different bacteria-affecting responses to ICB therapy, possibly due to the differences in cancer types and the environments. In order to reach a clear consensus on the definition of good and bad bacteria in cancer immunotherapy, multicenter cohorts around the world need to be studied.

As mentioned above, using a colorectal cancer model, Geller et al. [30] found that bacteria metabolize the gemcitabine (2′,2′-difluorodeoxycytidine), which is a common chemotherapeutic drug for PDAC to an inactive form, 2′,2′-difluorodeoxyuridine. A long isoform of the bacterial enzyme cytidine deaminase (CDDL) was expressed primarily in gamma *Proteobacteria*, which was involved in gemcitabine resistance in tumors, and administration of the antibiotic abrogated the gemcitabine resistance. They reported that 76% (86/113) of PDACs were mainly positive for gamma *Proteobacteria*.

### 5.3. Key Challenges and Limitations in Experiments of Microbiomes in PDAC

To elucidate the role of the microbiome in PDAC treatment, mouse models are commonly used. Fecal microbiota transplantation (FMT) was performed in immunocompetent mice after antibiotic treatment to analyze the role of fecal microbiota of interest in tumor formation [114,115,117]. Genetically engineered mouse models and orthotopic transplantation mouse models enable the analysis of the role of the microbiome in the formation of immunosuppressive TME including T-cell infiltration in tumors. However, the human microbiome not only differs significantly from that of the mouse but also among humans [118,119]. The gut microbiome is altered by a variety of factors including nutrition, antibiotics, probiotics, geography, and age [120], and the gut bacteria is considered to be influenced by the environment much more than ethnicity, race, and genetic background [121,122]. In addition, the rodent gut microbiota varies from laboratory to laboratory and source to source [123,124], creating problems with reproducibility in preclinical studies. To solve these problems, studies using the aforementioned mouse models are useful; the crosstalk between the microbiome and immunity has been well-studied [114,115,117] and the microbiome has been found to play an important role in cancer.

### 5.4. Antibiotic Treatment and Bacterial Transplantation Therapy in PDAC

The impact of microbiome ablation on PDAC development has been tested by antibiotic therapy as well as the activation of pattern recognition receptors (PRRs), such as TLR4 [101], TLR7 [125], Dectin [126], the NLRP3 inflammasome [127] in immune cells, and TLR9 in pancreatic stellate cells promotes carcinogenesis in the pancreas [100], which was abrogated by oral antibiotics. Thomas et al. [128] reported that Kras^G12D/+^;PTEN^lox/+^ mice depleted of microbes via antibiotics had a reduced percentage of poorly differentiated tumors compared to Kras^G12D/+^;PTEN^lox/+^ mice with intact microbes. Sethi et al. [129] reported that eradicating the microbiome with oral antibiotics significantly reduced the tumor volume in PDAC models as well as melanoma and colorectal cancer in an adaptive immune-dependent manner. The authors showed that decreasing the gut bacteria significantly increased interferon gamma-producing T cells and decreased in interleukin 17A- and interleukin 10-producing T cells, suggesting that modulation of the gut bacteria may be a new immunotherapy strategy. Pushalkar et al. [15] reported that the pancreases of PDAC patients contain more *Proteobacteria*, *Euyarchaeota*, and *Synergistetes* than normal pancreases, and ablation of the microbiota showed an enhanced effect of ICB in PDAC. The authors reported that the removal of the microbiome suppresses the development of both pre-invasive and invasive PDAC but the transfer of bacteria from tumor-bearing hosts promotes tumors. Bacterial removal was associated with immunogenic reprogramming of the TME in PDAC, including a decrease in MDSCs and an increase in M1 macrophage, which promoted Th1 differentiation of CD4+ T cells and activation of CD8+ T cells. These data suggest that endogenous microbes promote the immunosuppressive TME of PDAC and that microbial ablation is a promising approach to inhibit the progression of PDAC. However, a contrary effect of microbial ablation has been reported in other types of cancer [130,131,132], suggesting that the antibiotic effects are context-dependent. A phase I trial examining the effects of microbiome ablation in human PDAC may answer important questions about the role of specific microbiota in anti-tumor immunity (NCT03891979xii). Patients with resectable PDAC receive antibiotics and ICB therapy for 4 weeks before surgical resection. To reveal the effect of microbiome regulation in the immunotherapy of human PDAC, analysis of tumor tissue provides clues for markers of immune cell activation. Using treatment-naïve primary tumors enables answering how the removal of the microbiome contributes to changes in stromal and immune cell activity. Further studies are needed to evaluate the effect of antibacterial treatment on the tumor microenvironment and the efficacy of combining drugs including ICB therapies.

Riquelme et al. [14] performed human fecal microbiota transplants from PDAC patients, PDAC survivors, and healthy controls to tumor-bearing mice to evaluate the role of the gut microbiome in shaping the tumor microbiome, the immune system, and PDAC progression. The authors showed that gut or tumor microbiomes from PDAC survivors induced an antitumor response and enhanced the infiltration of CD8+ T cell in tumor-bearing mice, which was due to the decreased tumor infiltration of regulatory T cells (Tregs). These data suggested the causal role of the microbiome on the tumor microenvironment. As such, bacterial transplantation is a potential strategy for PDAC treatment. The efficacy of oral administration of a single or consortium of bacterial species as well as engineered non-pathogenic bacteria have been reported in other types of cancer [133,134,135]. Sivan et al. [133] reported that oral administration of Bifidobacterium alone improved antitumor immunity due to enhanced CD8+ T cell priming by augmented dendritic cell function in melanoma. Chowdhury et al. [135] engineered a non-pathogenic Escherichia coli strain which released an encoded nanobody antagonist of CD47 within the tumor microenvironment, stimulating the tumor-infiltrating T cells and systemic tumor-antigen-specific immune responses. In PDAC, these novel technologies are largely unexplored. Further analyses in preclinical and clinical studies are needed to test the efficacy of bacterio-therapies in PDAC.

## 6. Conclusions and Future Perspectives

The role of the microbiome in PDAC is summarized in Figure 3. Many observational studies have revealed the association of the oral, gut, and intratumor microbiomes with human PDAC. Intensive analyses using mouse models, including immunocompetent GEMMs and transplantation models, suggested that the microbiome has a systemic effect by bacterial translocation and systemic inflammation, etc. In addition, the microbiome may affect the composition of the tumor microenvironment via the immune response and generate an immunosuppressive environment. Furthermore, metabolites derived from the microbiome could affect the chemoresistance. These effects from the microbiome can be treated with antibiotics or bacterial transplantation.

In pancreatic precursors, such as IPMN and chronic pancreatitis, the microbiome may play a role in a pro-tumorigenic effect including inflammation and immunomodulation. Whether differences in the microbiome in pancreatic cancer precursors are a cause or a consequence remains unclear. Regarding its role as a biomarker, comprehensive analyses of multi-omics, including the microbiome, are expected to detect PDAC at an early stage. Therapeutic strategies targeting PDAC-associated microbiomes include the elimination of pro-tumorigenic bacteria with antibiotics or transplantation of bacteria to induce an immune-activated tumor microenvironment. The microbiome could also be a biomarker for the prediction of an immunogenic tumor microenvironment and immune-targeted therapies. A number of clinical trials are underway, and we look forward to the information from them as well as further studies for microbiome-based medicine.

## Figures and Tables

**Figure 1 cancers-14-04479-f001:**
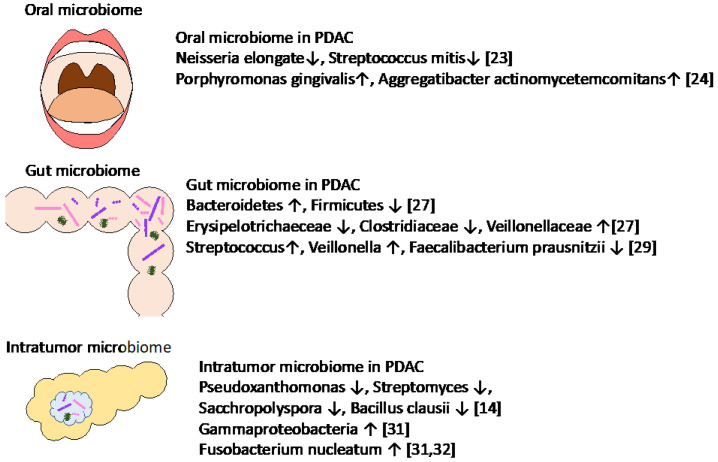
Specific microbiota associated with PDAC.

**Figure 2 cancers-14-04479-f002:**
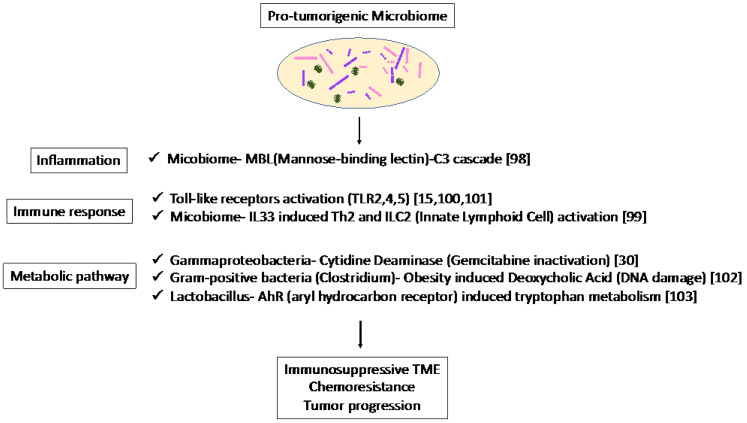
Specific microbiota and associated metabolic and biochemical pathways.

**Figure 3 cancers-14-04479-f003:**
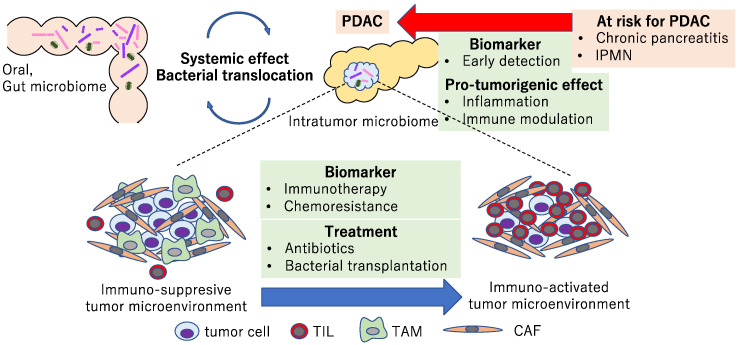
Diagram showing the association and potential roles of microbiomes with pancreatic precursors and PDAC. The microbiome may be a biomarker used to detect PDAC at an early stage and plays a pro-tumorigenic role via inflammation and immunomodulation. Microbiome-targeted therapies could be new therapeutic strategies in PDAC.

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
