# Peer review of "The Role of the Microbiome in Pancreatic Cancer"

_cancers, 2022, doi:10.3390/cancers14184479_

Round 1

Reviewer 1 Report

Title of the review paper can be changed

Subheading in the entire manuscript are redundant and not clear

Association and role of microbiome with PDAC- The descriptions are not elaborate

Transplantation of fecal pellets in immunocompetent mice with cancer to see how it shapes the TME and immune microenvironment- how are the authors trying to implicate these findings is not clear. 

Figure representation of potential role of microbiome in early detection of PDAC is not evident. The message that the authors are trying to convey not clear from this figure. The schematic should be well described in the legend.

This review can be more precise. 

Author Response

We appreciate that the Reviewer recognizes the potential importance of our findings and we attempt to  address the concerns below. 

Title of the review paper can be changed

Subheading in the entire manuscript are redundant and not clear

  • We regret that the subheadings in the entire manuscript were redundant and not clear. We think the title of the review could be the same, however, we have corrected redundant subheadings as the Reviewer pointed out. We have changed the title of sub-section2.1 and divided the sub-section2.1 into 2 sub-sections as “2-1. Association of Oral and Gut Microbiomes in PDAC” and “2.2. Association of Intra-tumor Microbiomes in PDAC” so that the sub-sections could be more associated with the title of the section.
  • We have changed the title of third section and sub-sections to “Association of Microbiomes with pancreatic diseases at high risk for PDAC”, "3.1. Association of Microbiomes with Chronic Pancreatitis”, and “3.2. Association of Microbiomes with Precursor Diseases of PDAC” because we described the association of microbiome with chronic pancreatitis and pancreatic precursors rather than the role of microbiome.
  • To address the redundancy, we have moved the sub-section “2.2. Mechanisms of Role of Microbiomes in PDAC” to “4. Mechanisms of Role of Microbiomes in PDAC” and combined this sub-section with the original “ Role of Microbiomes in PDAC development and TME” section. We made new sub-sections as “4.1. Association of Microbiomes with Molecular Subtypes of cancer cell” and “4.2. Role of Microbiomes in TME”. The contents in the original “2.2Mechanims of Role of Microbiome” section was included in “4.2. Role of Microbiomes in TME”.

Association and role of microbiome with PDAC- The descriptions are not elaborate

  • We appreciate the Reviewer’s point. For clarity, we generated new Figure1 to show the association of oral, gut, and intra-tumor microbiome with PDAC and Figure2 to show the mechanistic roles of microbiome in PDAC.

Transplantation of fecal pellets in immunocompetent mice with cancer to see how it shapes the TME and immune microenvironment- how are the authors trying to implicate these findings is not clear.

  • We appreciate the Reviewer’s points and we have corrected this part. We have added the sentence “These data suggested the causal role of microbiome on tumor microenvironment.” to emphasize the point that the adding microbiome could alter the tumor microenvironment rather than the association between microbiome and tumor microenvironment.

Figure representation of potential role of microbiome in early detection of PDAC is not evident. The message that the authors are trying to convey not clear from this figure. The schematic should be well described in the legend.

  • We appreciated the Reviewer’s comment and we have described our message in the legend and in the section” Conclusions and Future perspectives”

This review can be more precise.

  • We appreciated the Reviewer’s comment and we have revised the manuscript to be more precise as described above.

Reviewer 2 Report

In this manuscript, Miyabayashi et. al. presents a correlation between Pancreatic cancer and oral, gut, and intra-tumoral microbiome. This is an important topic and many reviews have presented an overview of this subject recently (PMID-34359684, 34455517, 31109338, 34884327 etc). The manuscript needs an overall significant improvement.

1-    The review has flaws in structural organization

·      The sub-sections within “2. Association of Oral, Gut, and Intratumor Microbiomes with PDAC” should be more associated with the title of the section.

·      The section “4. Role of Microbiomes in PDAC development and TME” its subsections are wordy and hardly emphasizes the microbiome.

·      Conclusions and Future perspectives section is poorly written.

2-    There is redundancy in the content of the review and sometimes the exact same line was repeated for instance, “The microbiome is now known to be associated with cancer development and progression in many types of cancer” has been repeated in lines 58-59 and 80-81.

3-    Macrophages and DCs are present in tumor stroma, but they are immune cells, whereas fibroblasts are stromal cells. The authors should draft this statement more precisely “Tumor-infiltrating T cells play an important role in eliminating tumor cells, and these components are regulated by other types of stromal cells such as fibroblasts, macrophages, and dendritic cells.

4-    Reference missing for “Lipopolysaccharide and TLR4 ligation induce a dendritic cell dependent immune response in the pancreas and increase pancreatic tumorigenesis.”  

5-    Line 371 typo “PADC”

Author Response

We appreciate that the Reviewer recognizes the potential importance of our manuscript and we attempt to  address the Reviewer’s concerns below.

1-    The review has flaws in structural organization

  • The sub-sections within “2. Association of Oral, Gut, and Intratumor Microbiomes with PDAC” should be more associated with the title of the section.
  • We appreciated the Reviewer’s comments. In relation to the comment from the Reviewer1, we have changed the titles of section2 and sub-sections as“2. Association of Oral, Gut, and Intratumor Microbiomes with PDAC”, “2-1. Association of Oral and Gut Microbiomes in PDAC” and “2.2. Association of Intra-tumor Microbiomes in PDAC”.

  • The section “4. Role of Microbiomes in PDAC development and TME” its subsections are wordy and hardly emphasizes the microbiome.
  • We thank the Reviewer pointing this out and have corrected it. To address the redundancy, we have moved the sub-section “2.2. Mechanisms of Role of Microbiomes in PDAC” to “4. Mechanisms of Role of Microbiomes in PDAC”and merged this sub-section with the original “ Role of Microbiomes in PDAC development and TME” section. We have created new sub-sections as “4.1. Association of Microbiomes with Molecular Subtypes of cancer cell” and “4.2. Role of Microbiomes in TME”. The contents in the original “2.2Mechanims of Role of Microbiome” section was included in “4.2. Role of Microbiomes in TME”.

  • Conclusions and Future perspectives section is poorly written.
  • We thank the Reviewer’s comment and have corrected it.

2-    There is redundancy in the content of the review and sometimes the exact same line was repeated for instance, “The microbiome is now known to be associated with cancer development and progression in many types of cancer” has been repeated in lines 58-59 and 80-81.

  • We thank the Reviewer pointing this out and have corrected it. To address the redundancy, we mergeed the sub-section “2.2. Mechanisms of Role of Microbiomes in PDAC” with “ Role of Microbiomes in PDAC development and TME” section as mentioned above.

3-    Macrophages and DCs are present in tumor stroma, but they are immune cells, whereas fibroblasts are stromal cells. The authors should draft this statement more precisely “Tumor-infiltrating T cells play an important role in eliminating tumor cells, and these components are regulated by other types of stromal cells such as fibroblasts, macrophages, and dendritic cells.”

  • We thank the Reviewer pointing this out and have changed the sentence to “Tumor-infiltrating T cells play an important role in eliminating tumor cells, and these components are regulated by other types of cells in tumor microenvironment (TME) such as fibroblasts, macrophages, and dendritic cells”.

4-    Reference missing for “Lipopolysaccharide and TLR4 ligation induce a dendritic cell dependent immune response in the pancreas and increase pancreatic tumorigenesis.” 

  • We thank the Reviewer pointing this out and have corrected it.

5-    Line 371 typo “PADC”

  • We thank the Reviewer pointing this out and this part has been removed to address the redundancy.

Reviewer 3 Report

The manuscript ID entitled "Pancreatic ductal adenocarcinoma (PDAC) is one of the most lethal malignancies, with little improvement in outcomes over the decades, although the molecular and phenotypic characterization of PDAC has contributed to advances in tailored therapies. PDAC is characterized by dense stroma surrounding tumor cells, which limits the efficacy of treatment due to creating a physical barrier and immunosuppressive environment. Emerging evidence regarding the microbiome in PDAC implies its potential role in the initiation and progression of PDAC. However, the underlying mechanisms of how the microbiome affects the local tumor microenvironment (TME), as well as the systemic immune system, have not been elucidated in PDAC. In addition, therapeutic strategies based on the microbiome have not been established. In this review, we summarize the current evidence regarding the role of the microbiome in the development of PDAC and discuss a possible role of the microbiome in the early detection of PDAC in relation to premalignant pancreatic diseases such as chronic pancreatitis and intraductal papillary mucinous neoplasm (IPMN). In addition, we discuss the potential role of the microbiome in the treatment of PDAC, especially in immunotherapy, although biomarkers to predict the efficacy of immunotherapy in PDAC is still unknown. A comprehensive understanding of tumor-associated immune responses, including those involving the microbiome, holds promise for new treatments in PDAC.

I appreciate the authors for the great effort in this good review. However, the following changes are required to improve the quality of the manuscript

The author may differentiate the association of microbiomes in the oral, gut, and intratumor with PDAC through scientific diagrams or tables to attract the scientific audience. 

The author may include key microbes and associated metabolic and biochemical pathways as a diagram to define "Mechanisms of Role of Microbiomes in PDAC"

What are the key challenges and limitations of the microbiome as a Biomarker for Immunotherapy and Chemotherapy in PDAC?

Author Response

We thank the Reviewer for the helpful feedback.  We have addressed the Reviewer’s suggestions in the bullet points below.

The author may differentiate the association of microbiomes in the oral, gut, and intratumor with PDAC through scientific diagrams or tables to attract the scientific audience.

  • We appreciate the Reviewer’s point and we generated new Figure1 to show the association of oral, gut, and intra-tumor microbiome with PDAC.

The author may include key microbes and associated metabolic and biochemical pathways as a diagram to define "Mechanisms of Role of Microbiomes in PDAC"

  • We appreciate the Reviewer’s comment and we generated new Figure2 to show the key microbiomes and associated metabolic and biochemical pathways in PDAC.

What are the key challenges and limitations of the microbiome as a Biomarker for Immunotherapy and Chemotherapy in PDAC?

  • We appreciate the Reviewer’s point and we have changed the title of the sub-section 5.3. as “Key challenges and limitations in Experiments of Microbiomes in PDAC” because key challenges and limitations of microbiome experiments are described in this sub-section. We think that it is challenging to assess causal and diagnostic roles of microbiome in human PDAC because microbiomes in mice and human are different. Many studies using mouse models have been conducted to assess the role of microbiome in PDAC. Further analyses in clinical studies are needed.

Round 2

Reviewer 1 Report

The edits suggested have been met.

Author Response

We are pleased that the reviewer feels the revisions have improved our manuscript.

Reviewer 2 Report

The Authors have addressed all of my concerns with the original manuscript.  However, the revised manuscript  require extensive editing of English.

Author Response

We are pleased that the reviewer feels the revisions have improved our manuscript.

As the reviewer considered that English editing was necessary, we have requested an English editing service to improve the English language.